# Factors influencing the effectiveness of nature-based interventions (NBIs) aimed at improving mental health and wellbeing: Protocol of an umbrella review

Topaz Shrestha[1]*, Cheryl Voon Yi Chi[1], Marica Cassarino[1,2], Sarah Foley[1], Zelda Di Blasi[1]

1 School of Applied Psychology, University College Cork, Cork, Ireland, 2 Environmental Research Institute, University College Cork, Cork, Ireland

☉ These authors contributed equally to this work.

* 115441352@umail.ucc.ie

## Abstract

Several systematic reviews support the use of nature–based interventions (NBIs) as a mechanism of enhancing mental health and wellbeing. However, the available evidence for the effectiveness of these interventions is fragmentary and mixed. The heterogeneity of existing evidence and significant fragmentation of knowledge within the field make it difficult to draw firm conclusions regarding the effectiveness of NBIs. This mixed method umbrella review aims to synthesise evidence on the effectiveness of nature–based interventions through a summative review of existing published systematic reviews and meta-analyses. A systematic search in PsycINFO, PubMed, Greenfile, Web of Science, Embase, Scopus, Academic Search Complete (EBSCO), Environment Complete (EBSCO), Cochrane Library, CINAHL, Health Policy Reference Centre and Google Scholar will be performed from inception to present. The search strategy will aim to find published systematic reviews of nature–based interventions (NBIs) where improving health and wellbeing is an explicit goal. This is a mixed method review, and systematic reviews with both quantitative and qualitative data synthesis will be considered. Two authors will independently perform the literature search, record screening, data extraction, and quality assessment of each included systematic review and meta-analysis. The individual qualitative and quantitative syntheses will be conducted in parallel and combined in an overarching narrative synthesis. The quantitative evidence will be used to assess the strength and direction of the effect of nature–based interventions on mental health and wellbeing outcomes. Evidence drawn from qualitative studies will be analysed and synthesised to understand the various pathways to engagement, involvement process and experiential factors that may mediate experiences. The risk of bias of the systematic reviews will be assessed using a 16-item Assessment of Multiple Systematic Reviews 2 (AMSTAR2) checklist.

**Trail registration:** This review is registered on PROSPERO (CRD42022329179).

**Data Availability Statement:** This is a protocol for a systematic umbrella review, thus no empirical

data has been collected to prepare this manuscript. No datasets were generated or analysed during the current study.

**Funding:** The author(s) received no specific funding for this work.

**Competing interests:** The authors have declared that no competing interests exist.

# Introduction

## Disconnection from nature

Connecting with nature is an important element of many people's lives, and a substantial body of research supports nature's restorative influence on our mental health and wellbeing [1–3]. More recently, the benefits of nature–based interactions are becoming increasingly acknowledged across disciplines from Positive Psychology and Urban Planning to Medicine and Public Health. This research demonstrates a consistent positive trend between engagement with nature and improved physical and mental health outcomes [4, 5]. Therefore, it is of significant concern that urbanisation, environmental degradation, and the challenges of modern living are leading to a reduction in engagement with the natural environment. A presiding narrative in industrialised nations is that modern-urbanized lifestyles have diminished healthy human relationships with natural environments resulting in a multitude of health issues and reduced wellbeing [4, 6]. Many of us seem physically and psychologically disconnected from nature, which has implications for the wellbeing of the environment and individuals [7]. While long-acknowledged as practices across cultures, nature–based therapeutic interventions have grown substantially in number and type in recent years [1, 8]. Western science is beginning to realise what Indigenous cultures have always known–engagement with the natural environment can support, enhance, and restore our health and wellbeing [5].

## Nature-based interventions

There is growing interdisciplinary interest in the potential for 'nature–based interventions' (NBIs) to promote and restore mental health and wellbeing. NBIs can facilitate change through a relatively structured promotion of nature–based experiences. Although a generally accepted definition is lacking, NBIs can be defined as intentional programmes, activities or strategies that aim to engage people in nature–based experiences with the specific objective of enhancing health and wellbeing [9, 10]. NBIs are deliberate therapeutic processes recognising nature–human kinship [11]. These interventions can be broadly categorised into those that change the environment in which people live, learn, work, recreate and heal (for example, the provision of parks in cities or gardens in hospitals) and those that alter behaviour (for example, engaging people through organized programmes such as wilderness therapy) [9]. Considerable variety exists in practice, for example, biophilic design in urban settings, the incorporation of green and blue spaces in cities, the development of community gardens [12, 13]. Additionally, there are interventions which revolve around more active engagement with nature. For example, green exercise programmes, ecotherapy [3, 10], therapeutic horticulture, sea swimming initiatives, forest bathing and expedition-based wilderness programmes [5, 8]. These interventions can be centred around green space, blue space, or an amalgamation of both. Greenspace is habitually comprised of vegetation and is associated with natural elements. There are two interpretations of greenspace. Firstly, the interpretation that greenspace refers to areas of vegetation in a landscape, such as forests and wilderness areas, gardens and backyards, street trees and parks, farmland, geological formations, coastal regions, and food crops. This interpretation encompasses the overarching concept of nature or natural areas in general. The second interpretation focuses on urban vegetation, including parks, gardens, urban forests, and farms – usually relating to a vegetated variation of open space [12]. Blue space can be defined as 'all visible, outdoor, natural surface waters with potential for the promotion of human health and wellbeing', e.g., rivers, lakes, coasts, sea, etc. (p. 52). Research has highlighted the specific potential for freshwater, coastal and marine ecosystems to promote and restore mental health and wellbeing [2, 11]. It is evident that there is considerable overlap between blue and green

spaces. However, these offer very different sensory experiences and are used in varying ways with distinct health outcomes and benefits that are often overlooked and remain poorly understood.

Many existing reviews of NBIs define nature exposure using metrics such as the amount of green or blue space present in any given area (e.g., number of parks with access to greenery, lakes, etc.) [1, 14, 15]. An inherent limitation of these metrics is that they assume exposure revolves around geographic proximity without considering whether nearby nature was used, of good quality, or whether it was accessible (e.g., near a busy road crossing) [16, 17]. This has resulted in a call for researchers to broaden their definition of nature exposure and investigate different types of natural settings and their characteristics [1, 14, 18]. To address this, the current review focused on NBIs where 'nature-based' encompasses what Bloomfield [3] refers to as "time spent outside in places defined as rich in natural beauty and biodiversity" (p. 82). This includes both biodiverse, unregulated, wild nature lacking human involvement [3] and publicly accessible, managed urban green spaces or blues paces (e.g., parks, gardens/allotments, and artificial lakes/reservoirs) [19]. Understanding what NBIs are available and the various factors influencing the effectiveness of these interventions is necessary if we are to gain a clear picture of the current state of the research.

## Meta perspective on the rise of nature-based interventions

It is important to acknowledge that nature-based interventions are part of a complex system affected by cultural, social, and political factors [1, 16]. The recent proliferation of NBIs may be linked to a broader cultural shift towards the 'natural' [3, 16]. Environmentalism has become topical and it is possible that NBIs are an 'eco-fad' that is trending right now. Over the past 30 years many individuals and organisations have begun to promote the human-centric approach to the protection of natural resources [7, 11]. This human-centric perspective on conservation highlights the interrelationship between humans and nature and our responsibility to protect the planet. The urgency of the climate crisis and subsequent pressure people feel could explain the interest in NBIs. Many of us are experiencing new emotions, such as climate anxiety and ecological guilt and these feelings of accountability may result in this mass movement towards nature-based solutions [7, 8, 11]. Additionally, the rise in interest in NBIs may be partially economically motivated, as western welfare states realise increasingly its cost and time-effectiveness [11–13, 16]. Lost in the rise of these eco-fads is an objective appraisal of whether these actions have a tangible impact on human and environmental health, or merely pacifying our ecological guilt. Environmental policies are designed primarily to construct a green image—rather than deliver results. Given the complex social and economic criticism of NBI's as eco-fads, we acknowledge this dimension and are committed to providing a critical lens to NBIs.

## Nature-based interventions, mental health, and wellbeing

Globally, the growing interest in the restorative potential of NBIs, within healthcare, seems to be driven by a global mental health crisis and the rise of non-communicable diseases [1, 20]. The issue of mental health and wellbeing is particularly pertinent with rising suicide rates and lack of funding for services highlighted internationally [21]. Moreover, evidence shows a significant gravitation towards natural environments during the COVID-19 pandemic and that this increased engagement with nature may have buffered the negative mental and behavioural impacts of recurrent lockdowns [22]. Public health administrations are beginning to acknowledge the significance of proximity to, and engagement with, natural environments 'as an upstream health promotion intervention for populations' [23]. The recognition of the value of

nature and place as a determinant of mental health and wellbeing presents a crucial opportunity for struggling healthcare systems seeking new and cost-effective services [24].

## Reviewing the evidence on nature-based interventions

Several comparative studies, randomized controlled trials, observational studies, and subsequently systematic reviews and meta-analyses have been conducted to investigate the efficacy of nature–based interventions (NBIs) on mental health and wellbeing outcomes [2, 10, 25, 26]. Considering the extensive number of systematic reviews assessing NBIs, it is crucial to synthesise the findings of these reviews to consolidate the evidence and better inform science and practice [9]. Umbrella reviews are systematic overviews of systematic reviews and/or meta-analyses that can be used to provide a summary of the evidence from multiple research syntheses. Systematic reviews conducted with optimal methodological rigour can provide high-quality evidence informing further research and the development of effective policies. With the increased number of systematic reviews of NBIs available, a logical and necessary next step is to conduct an umbrella review of existing systematic reviews, allowing the findings of separate reviews and meta-analyses to be compared, subsequently providing decision makers in healthcare with an overall synthesis of the body of information available [27]. This is a rapidly growing field, and the recent interest in nature–based solutions (NBS) and the proliferation of 'nature–based interventions' (NBIs) is surpassing the policy and knowledge base. This has resulted in a general lack of understanding regarding the practical implementation of NBIs within public planning and policy and the factors influencing the effectiveness of these interventions [9, 28]. This can only limit the leveraging of natural environments to improve health and wellbeing outcomes, potentially resulting in ineffective and ill–targeted investment decisions. A higher-order or meta-level synthesis is required to make sense of this evidence. This will provide a broader picture of the types of interventions available, the specific mental health and wellbeing outcomes they impact upon, the drivers and barriers to using NBIs, and the methodological quality of the existing research. There is significant fragmentation of knowledge within the field and previous studies highlight evidence gaps concerning effectiveness of interventions [1, 9, 24]. The universal application of NBIs to different groups and the diversity of nature itself has led to significant heterogeneity of intervention designs [9]. There is a need for a comprehensive overview of existing evidence regarding the effectiveness of interventions, particularly given the plurality of types of interventions, delivery approaches, and patient groups [16]. Moreover, vague intervention descriptions and an absence of theoretical frameworks guiding NBI design has limited the critical appraisal of these interventions [27–31].

## Objectives of the review

The aim of this mixed method umbrella review is to synthesise the evidence on the effectiveness of NBIs aimed at enhancing mental health and wellbeing through a summative review of existing published systematic reviews and meta-analyses. This review aims to explore the drivers influencing the effectiveness of NBIs by exploring factors that encourage engagement in NBIs and potential barriers to involvement in these interventions. The decision to include both quantitative and qualitative evidence was based on the commitment to provide an extensive and accurate summary of the existing evidence of NBIs. It is expected that the quantitative evidence will be used to assess the strength and direction of the effect of NBIs on mental health and wellbeing outcomes, thereby providing insight into the effectiveness of interventions. In contrast, the qualitative studies will provide a more nuanced perspective of the factors influencing the effectiveness of nature–based interventions. Our qualitative analysis will attempt to capture the holistic experience of nature–based interventions for participants involved and

help to understand the experience and meaning of participation in nature–based interventions. By providing insight into pathways to engagement and potential factors which may mediate their experiences of these interventions. The mixed-methods synthesis of will provide a comprehensive overview of current evidence and will help to identify gaps in knowledge, potential quality needs and directions for future research. Further knowledge and communication about the effectiveness of interventions is likely to be a valuable precursor for their use [9].

The overall objective of this mixed method umbrella review is to synthesize the evidence on the effectiveness of nature–based interventions, aimed at enhancing mental health and wellbeing to explore the overarching research question: *What are the factors influencing the effectiveness of nature-based interventions*? Accordingly, our specific objectives were to identify: 1) what nature–based interventions (NBIs) are available, 2) what specific mental health and wellbeing outcomes might they achieve for whom, and 3) what are the factors that drive or limit the effectiveness of NBIs.

This umbrella review will provide a comprehensive synthesis of the growing evidence on NBIs to offer recommendations for future research, policymaking, and practice.

## Methods and analysis

### Protocol registration

The umbrella review will adhere to the predesigned protocol that we developed based on the Preferred Reporting Items for Systematic Review and Meta-Analysis Protocols (PRISMA-P) guidelines [32] (S1 Table). This project was registered with the International Prospective Register of Systematic Reviews (PROSPERO) (registration number CRD42022329179).

### Data sources and search strategies

We will conduct a comprehensive umbrella review of all available systematic reviews on the topic using the methodology described by Smith et al., which outlines the methodology for conducting reviews of systematic reviews of healthcare interventions [31]. We will adopt the Joanna Briggs Institute (JBI) methodology for umbrella reviews, which provide further guidelines specific for synthesizing the findings from multiple reviews [29]. The systematic overview resulting from the conduct of an umbrella review is useful to explore whether the evidence base around a topic is consistent or contradictory, and to examine the reasons for the findings [29, 33].

A systematic search of the following twelve databases will be completed: PsycINFO, PubMed, Greenfile, Web of Science, Embase, Scopus, Academic Search Complete (EBSCO), Environment Complete (EBSCO), Cochrane Library, CINAHL, Health Policy Reference Centre and Google Scholar. No date limit will be placed on the search. The search strategy will aim to find published systematic reviews of nature–based interventions (NBIs) where improving health and wellbeing is an explicit goal. Our search strategy will be comprised of three elements. Search terms relating to (i) nature–based interventions/green or blue spaces and (ii) mental health and wellbeing outcomes will be combined with (iii) systematic review OR meta-analysis and searched for in title, abstract, and keywords. We will search databases using a set of search query including keywords and Boolean operators to retrieve the relevant literature as per the objective of this review. The search strategy consists of keywords related to the natural environment, mental health, and systematic review. The selection of search terms was based on existing theories and research defining nature–based interventions as well as initial preliminary searches for the umbrella review. Each search term will be applied twice—initially by itself, then paired with the term "systematic review" to reduce the number of returns on some of the searches, with additional searches using hyphenated variants where appropriate. The search terms for nature–based interventions/green or blue spaces and health/wellbeing

outcomes will be combined with the Boolean AND; within each group the Boolean OR will be used. Aiming for as complete coverage as possible, the search may be widened, beyond the protocol, by scanning identified articles' bibliographies and "snowballing." The detailed search strategy, which was developed by the full research team in consultation with a Faculty Librarian, is available in S2 Table. The results of the search will be fully reported in the final study and presented in a flowchart following the PRISMA guidelines.

Our inclusion criteria are based on the Cochrane criteria for what constitutes a systematic review as well as the AMSTAR 2 tool for the quality assessment of systematic reviews, in order to incorporate only high-quality systematic reviews [34]. The AMSTAR 2 domains will be used as indicators of eligibility for our study. We will include all systematic reviews and meta-analyses that investigate the impact of nature–based interventions (NBIs) on mental health and wellbeing. The search will be limited to peer-reviewed studies published in English and results will be filtered, by study type, to include solely systematic reviews. Our decision to include only peer-reviewed studies is based on our commitment to provide a synthesis of high-quality evidence which has gone through a meticulous and rigorous review process [27, 29]. Additionally, our decision to include only studies published in English is the result of limited resources and the language constraints of our review team. Including studies published in non-English languages would have increased resource challenges in relation to time, costs, and expertise in non-English languages. This umbrella review will include systematic reviews, with or without meta-analysis, which review any type of nature–based intervention (NBI). Unpublished grey literature will not be included due to the fact that the focus of this review is on systematic reviews and meta-analyses with quality assessments, which are typically found in academic peer-reviewed publications. Inclusion criteria will be restricted to studies with defined search terms, inclusion criteria and quality assessment.

The above-mentioned criteria are fundamental components of a high-quality systematic review [1]. Systematic reviews which examine both randomized controlled trials (RCT) and non-randomized controlled trials and observational studies (which do not have a control group) will be included in this overview. The rationale behind this decision is that the field largely consists of non-randomized trials, and excluding systematic reviews which include non-RCT studies may result in an incomplete synthesis of findings [35].

## Data selection

A PRISMA flow chart will be developed to record the screening and selection of studies. Once all records from our search are collected, EndNote 20 software (Thomson Reuters, Toronto, Ontario, Canada) will be used to remove duplicates and screen literature. A 2-stage screening process will be completed independently by two researchers (TS and CVYC); the first screening stage will consider titles and abstracts, while full-texts will be checked in the second stage. Interrater reliability (IRR) will be reported at all three stages of screening and data extraction to ensure consistency and clarity [36]. Any disagreements will be resolved by discussion or by the involvement of a third reviewer (ZDB) until consensus is reached. When titles and abstracts are insufficient to determine whether to include or exclude reviews, we will download full texts to determine eligibility. Based on the umbrella review methodology, when numerous systematic reviews provide duplicated datasets for the same comparison, the systematic review with the greatest number of studies providing study-level effect estimates will be retained for further analysis [37]. The following are the detailed inclusion criteria:

## Participants

There are no age or gender restrictions for participants. Children, adolescents, and adults with or without mental and/or physical health problems. The routes to participation (e.g.,

motivations and barriers) will be considered throughout analysis to further understand how nature–based interventions could influence health and wellbeing of participants and in what contexts. It is anticipated that the qualitative evidence will provide insight into the routes to participation in nature–based interventions.

## Interventions

In this umbrella review we will include any systematic review focused on real nature–based interventions (NBIs)/exposure to green and blue spaces. Real nature is defined as a broad range of green landscapes in the indoor and outdoor environment. This includes green spaces (e.g., botanic garden, or tree canopy), indoor nature (e.g., potted plants, green walls, or flowers), or real nature views (e.g., window views) [38]. For this study, NBIs are defined as programmes, activities or strategies that aim to engage people in nature–based experiences with the specific intention of improving health and wellbeing outcomes [9].

In order to adopt a comprehensive definition of nature exposure which considers different types of natural settings and their characteristics [1, 14, 18], the current review focused on NBIs which included exposure to both 'wild'/unregulated natural environments which have minimal human involvement and managed/man-made green and blue spaces such as urban parks, ponds, garden allotments etc. Additionally, we included NBIs which focused on biophilic design for example, green walls in cities and the overall incorporation of nature into infrastructure. All included studies must encompass NBIs which integrate explicit and purposeful nature contact, incorporating blue or green space through direct nature exposure to an authentic natural setting (e.g., walking in nature/ being in a park, etc.). We will exclude interventions that examined the effects of artificial nature, virtual/simulated nature, animal therapy, animal interventions, fish tanks, or nature sounds. The justification of this revolves around our focus on 'real nature' [38] and our conception of NBIs where 'nature-based' encompasses what Bloomfield [3] refers to as "time spent outside in places defined as rich in natural beauty and/or biodiversity" (p. 82).

## Outcomes

All systematic reviews which assess the mental health and wellbeing impacts experienced by individuals following active participation in a nature–based interventions will be included. Mental health, as defined by the World Health Organization (WHO), is "a state of wellbeing in which the individual realizes his or her own abilities, can cope with the normal stresses of life, can work productively and fruitfully, and is able to make a contribution to his or her community" [39]. Wellbeing encompasses positive emotions and mood, the absence of negative emotions, satisfaction with life, fulfilment, and positive functioning [40, 41]. All included interventions must have the promotion of mental health and wellbeing outcomes as an explicit goal (i.e., programmes that solely aim to connect people with nature without the objective of also delivering health and wellbeing benefits will be excluded).

**Quantitative research.**   Includable primary outcomes will include any recognised measure of mental health and wellbeing assessed using self-reported and objective measures. Outcomes can be defined as the psychological effects of NBIs related to mental health and wellbeing (e.g., life satisfaction, quality of life, vitality, stress, anxiety, exhaustion, burnout, and depression). The outcomes can be categorized as: (i) mental health indices, (ii) restoration and recovery, (iii) executive functioning/cognitive ability, (iv) work and life satisfaction, and (v) psychophysiological indicators of psychological wellbeing (e.g., cortisol levels).

**Qualitative research.**   Includable qualitative evidence will be in the form of themes, metaphors and concepts relating to the meaning, experience and perceived effects of nature–based

interventions and any factors that help or hinder their success, e.g., direct quotes, and author analysis of qualitative findings.

## Data collection and verification

We will develop a standardized form for extracting data from each systematic review. The ad hoc data extraction sheet will be developed and piloted prior to data collection and will be used to ensure a controlled analysis and data retrieval. Two authors will collect the variables listed below and cross-check the accuracy of the data. A plan for data extraction is shown in Table 1 below:

## Critical appraisal

Methodological quality of the included systematic reviews will be assessed by two independent researchers using the Assessment of Multiple Systematic Reviews 2 (AMSTAR2, an updated version of AMSTAR) tool, a 16-item checklist used to critically rate the quality of an individual systematic review as high, moderate, low and critically low based on the total score of the AMSTAR2 [34]. The AMSTAR 2 tool has been updated to facilitate a more detailed assessment of systematic reviews that include both randomised and non-randomised studies of healthcare interventions. The risks of bias will be analysed in relation to the particular design, conduct, and synthesis of the systematic review. Risk of bias assessment will assess methods of randomization and intervention allocation. In the case of disagreements, a discussion will be conducted with a third reviewer to reach a consensus. In the case of insufficient or additional information, the study authors will be contacted.

## Data analysis

The strategy for data synthesis will consist of firstly extracting the quantitative and qualitative data from each review, which will be entered into the screening and data extraction table. Findings will be structured around a synthesis of the characteristics of included studies, the classification of interventions used, and the types of outcomes reported. A narrative synthesis will be used to present the potential factors influencing the effectiveness of NBIs. The umbrella review format will enable a unique form of evidence synthesis whereby the researchers can stand back and gain a comprehensive summary of the breadth of research on NBIs. The results will be reported descriptively in the text, and in tables. Visual techniques will be used to present the quantitative synthesis in a comprehensive format, for example a narrative approach

**Table 1. Data extraction table.**

| Data Extraction Item |
| --- |
| Author(s) |
| Year of publication |
| Country of origin |
| Number of articles included in systematic review |
| Search terms |
| Type/definition of intervention reviewed |
| Type/definition of mental health and wellbeing outcome(s) reviewed |
| Quality assessment |
| Quantitative findings–main findings and effect sizes |
| Qualitative findings–themes, metaphors and concepts relating to the meaning, experience, and perceived effects of nature–based interventions and any factors influencing effectiveness of NBIs e.g., direct quotes, and author analysis of qualitative findings |

table, indicating factors such as strength and direction/s of results and study quality) will be used to visually depict the trends in the results. Similarly, visual techniques will be used to illustrate the qualitative data and synthesis, e.g., graphs, tables, flow charts etc.

Quantitative studies will provide insight into the strength and direction of evidence of effect. However, we anticipate a limited scope for meta-analysis due to the likelihood that many individual studies will be included in more than one review, resulting in inaccurate statistical power and a risk for misleading results. Additionally, the heterogeneity of intervention type given the plurality of disciplinary origins of these interventions will further impede the potential for meta-analysis. As our review considers one type of "intervention" (nature–based), however of varying composition (e.g. ecotherapy, green infrastructure, or blue environments), and its effect on several different health outcomes, we consider the challenge of dissecting each included review, extracting the results from each individual study included, and the subsequent amalgamation of the results, to be of insubstantial value given the heterogeneity in the outcome measures and the unreliable accuracy of a pooled effect estimate [42]. A narrative synthesis approach will be used if the quantitative study design or outcomes are so heterogeneous to impede meta-analysis [43]. Where possible and appropriate, statistical heterogeneity will be assessed following the Cochrane 6.3 guidance [1]. Should trials of clinical relevance be included, an assessment of clinical heterogeneity, considering variability in participants, outcomes and characteristics of the intervention, will be assessed following guidance by Gagnier et al., [55].

Qualitative studies will be used to provide a more nuanced perspective of the factors influencing the effectiveness of nature–based interventions. It is anticipated that the qualitative analysis will capture the holistic experience of NBIs, for participants involved, and help to understand the experience and meaning of participation in NBIs, highlighting pathways to engagement and potential factors that mediate their experiences. Qualitative findings will be synthesised narratively by developing key themes related to enablers and barriers of effectiveness [43]. The synthesis will be sensitive to factors which may affect the impact on wellbeing, such as the demographics of participants, the context of the activities, and the implementation and specifics of the interventions.

## Overarching synthesis

The individual qualitative and quantitative syntheses will be conducted in parallel and then combined in an overarching narrative synthesis [43]. Narrative synthesis supports the contextualised integration of diverse forms of evidence helping researchers to understand the phenomenon of interest. This approach is especially suited to reviews of complex intervention effectiveness such as NBIs. If data permits, the analysis will be sensitive to impacts on different groups of people (e.g., age, those with mental ill health, those recovering from specific conditions or addictions). The qualitative evidence will also be used to explore those factors which help or hinder the successful development, implementation, and sustainability of the form of NBI for different groups of people. The combined narrative synthesis will be used to develop a conceptual model [44]. The model will be grounded in formulated on the synthesised results of both the qualitative and quantitative evidence. We are committed to practising reflexivity throughout the research process. We will consider the cultural, social, political, and ideological origins of your own perspectives throughout the study. We will critically examine our own role, assumptions, beliefs, pre-existing potential bias and impact on the data during all stages of the research process including: (a) formulation of the research questions (b) data collection and (c) data analysis. We hope that this reflexive engagement by multiple analysts will enhance the quality of this study [3].

Based on the evidence collated, we may decide to split the review into two separate sister papers, separated by quantitative and qualitative evidence—What do we know about NBI, parts I and II. We see significant merit in this distinction provided the evidence will support such a divide. We have currently completed full-text screening. The decision to separate evidence will be made when piloting our data extraction sheet. By piloting the extraction sheet gauge the potential scope to divide the review into quantitative and qualitative evidence, i.e., are there enough quantitative and qualitative reviews to warrant this divide and do the mixed reviews present their findings in a way that would facilitate the separation of quantitative and qualitative evidence.

## Discussion

By incorporating evidence from published systematic reviews and meta-analyses, we will provide a comprehensive overview of the factors influencing effectiveness of nature-based interventions (NBIs) which are aimed at enhancing mental health and wellbeing.

The recent and rapid proliferation of 'nature–based interventions' (NBIs), is surpassing the policy and knowledge base. This results in challenges in understanding and evaluating their tangible impact on the publics' mental health and wellbeing [5, 12, 24]. There is a need for further evidence regarding the effectiveness of interventions. While multiple interventions exist, all proposing engagement with nature as means of enhancing mental health and wellbeing, there is a dearth of guidance as to what NBIs are available and the drivers influencing their effectiveness [9]. This can only impede the leveraging of natural environments to improve mental health and wellbeing outcomes, potentially leading to ineffective and ill–targeted investment decisions. We postulate that this knowledge gap exists due to the diversity of intervention designs and therapeutic approaches. Moreover, the lack of financial prioritization allocated to cost-effective NBIs impedes the potential for such interventions to ameliorate health and wellbeing on a larger scale. With the rising prevalence of substandard mental health, and the established link between poor mental health and a myriad of other noncommunicable diseases, the general population bears a significant socioeconomic burden [45, 46]. We recognise that NBIs are part of a complex system influenced by social, cultural, and political factors. Subsequently, the pathways between health and nature are linked to health inequalities [1, 16, 47, 48]. It is often the most underprivileged, i.e., people with lower socioeconomic status, who benefit from access to and engagement with high-quality nature [12, 16, 49]. Nature–based interventions could be a cost and time-effective mechanism of enhancing wellbeing at a population level. However, a concerted and systematic effort is required to understand what factors influence the effectiveness of interventions [16]. Additionally, there is evidence that policy and decision makers around the world, who are interested in cost-effective health improvement programmes, are progressively supporting the promotion of 'nature–based solutions' (NBS) [1, 50, 51]. It is therefore timely that the evidence of effectiveness of nature–based interventions is reviewed in a systematic and rigorous manner.

With the increase in the number of systematic reviews conducted, a necessary next step to provide practitioners in healthcare with the evidence they need to successfully develop NBIs. An umbrella review was chosen to provide an overview of the evidence from multiple research syntheses through an overall examination of the body of systematic and analytic reviews. This form of evidence synthesis supports comparative analysis. This method allows us to collectively evaluate the state of the evidence in broad categories of research, which may make more sense in clinical practice rather than evaluating [them] one by one [27]. The umbrella reviews' most distinguishing feature is that only the highest level of evidence, namely other systematic reviews, and meta-analyses, are considered for inclusion [29, 31]. By synthesising high-level

evidence of the factors influencing the effectiveness of NBIs we will gain a comprehensive overview of the strengths and weaknesses of such interventions. Thus, supporting the implementation of interventions which are more targeted and subsequently more effective. Whilst there seems to be a considerable body of literature which has aimed to understand the potential mental health and wellbeing benefits of nature–based interventions, there are no available umbrella reviews that have addressed the factors influencing the effectiveness of interventions. Several linked reviews were identified but these were either limited in scope (e.g., focusing specifically on nature's role in psychotherapy [4], assessing exclusively built/urban natural environments [1, 52, 53], or don't focus explicitly on nature–based interventions e.g. exploring exposure to natural environments in general rather than intentional NBIs [2]. Moreover, to the best of our knowledge, this is the first mixed method umbrella review that summarizes the factors influencing the effectiveness of nature–based interventions (NBIs) thus providing insight into the practical application of NBIs within public planning and policy. The focus on mental health and wellbeing outcomes in wider contexts will provide a better understanding of potential approaches and pathways which are needed to support an evidenced-based knowledge of the importance of NBIs.

The empirical evidence relating to our research questions, both quantitative and qualitative, will be identified, appraised, and synthesized. Evidence drawn from qualitative studies will help to understand the diverse pathways to engagement and factors that mediate participants' experiences of NBIs. The aim of this umbrella review is not to repeat the searches, assessment of eligibility or risk of bias for the included reviews, but rather to provide an overall picture of findings for the specific phenomenon of NBIs. In contrast to systematic reviews or meta-analyses limited to one treatment comparison, an umbrella review can provide a broader picture of many intervention types [29, 54]. This is more effective in informing guidelines and clinical practice when all the management options must be considered. This umbrella review intends to provide a resource for decision–makers in government and other interested organisations by outlining potential interventions, the specific mental health and wellbeing outcomes they might achieve for whom, the drivers influencing the extent to which these interventions succeed, and the target beneficiaries. We expect that the findings of this review will provide a roadmap for decision–makers and support the integration of NBIs into public planning and policy.

There are some limitations inherent to our umbrella review. It is anticipated that the included systematic reviews will vary in their heterogeneity and quality. This is likely due to the diversity of intervention types, disciplinary origins of these interventions, delivery approaches, and patient groups for which they are being used [9, 16, 55, 56]. In addition, heterogeneity is driven by the breadth of the aims and uses of the interventions that will be potentially includable in the review, which will range from exposure to nature through to specific therapeutic interventions. As a result, we anticipate a limited scope for meta-analysis. We will use the AMSTAR 2 checklist to assess the risk of bias of each included study and address the concerns around the quality of included reviews. Given the resource constraints faced in undertaking this study, we assess only English literature. We acknowledge that this is a limitation of this work as it may privilege a particular perspective and reduces generalizability. Additionally, as mentioned in the introduction, the recent proliferation of NBIs may be linked to a cultural movement towards environmentalism and a desire to be perceived as environmentally conscious. We recognise that papers overtly critical of NBIs may be missing from the evidence base we collate. Where such a perspective is missing, we are committed to applying a more reflective and critical lens to the evidence to provide a comprehensive overview of the research.

Despite anticipated limitations, we believe that the result of this umbrella review will benefit practitioners, landscape, and urban design professionals, policymakers, and the general public.

A synthesis of the evidence including the methodological quality of the research will also be of great importance to researchers in this field.

## Supporting information

**S1 Table. PRISMA-P checklist.**
(DOC)

**S2 Table. Search strategy.**
(DOCX)

## Author Contributions

**Conceptualization:** Topaz Shrestha, Marica Cassarino, Sarah Foley, Zelda Di Blasi.

**Data curation:** Topaz Shrestha, Cheryl Voon Yi Chi, Marica Cassarino, Sarah Foley, Zelda Di Blasi.

**Formal analysis:** Topaz Shrestha, Cheryl Voon Yi Chi, Marica Cassarino, Sarah Foley, Zelda Di Blasi.

**Investigation:** Topaz Shrestha, Cheryl Voon Yi Chi, Marica Cassarino, Sarah Foley, Zelda Di Blasi.

**Methodology:** Topaz Shrestha, Cheryl Voon Yi Chi, Marica Cassarino, Sarah Foley, Zelda Di Blasi.

**Project administration:** Topaz Shrestha, Marica Cassarino, Sarah Foley, Zelda Di Blasi.

**Resources:** Topaz Shrestha, Marica Cassarino, Sarah Foley, Zelda Di Blasi.

**Software:** Topaz Shrestha, Cheryl Voon Yi Chi, Marica Cassarino, Sarah Foley, Zelda Di Blasi.

**Supervision:** Marica Cassarino, Sarah Foley, Zelda Di Blasi.

**Validation:** Topaz Shrestha, Cheryl Voon Yi Chi, Marica Cassarino, Sarah Foley, Zelda Di Blasi.

**Visualization:** Topaz Shrestha, Marica Cassarino, Sarah Foley, Zelda Di Blasi.

**Writing – original draft:** Topaz Shrestha.

**Writing – review & editing:** Topaz Shrestha, Cheryl Voon Yi Chi, Marica Cassarino, Sarah Foley, Zelda Di Blasi.

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
