## [Decision Letter · Decision Letter 0]

13 Oct 2022

PONE-D-22-21700Factors influencing the effectiveness of nature–based Interventions (NBIs) aimed at improving mental health and wellbeing: Protocol of an umbrella reviewPLOS ONE

Dear Dr. Shrestha,

Thank you for submitting your manuscript to PLOS ONE. After careful consideration, we feel that it has merit but does not fully meet PLOS ONE’s publication criteria as it currently stands. Therefore, we invite you to submit a revised version of the manuscript that addresses the points raised during the review process.

We look forward to receiving your revised manuscript.

Kind regards,

Nyanyiwe Masingi Mbeye, Ph.D

Academic Editor

PLOS ONE

Journal Requirements:

2. During our internal evaluation of the manuscript, we found significant text overlap throughout the whole mansucript between your submission and previous work  .

Please revise the manuscript to rephrase the duplicated text, cite your sources, and provide details as to how the current manuscript advances on previous work. Please note that further consideration is dependent on the submission of a manuscript that addresses these concerns about the overlap in text with published work.

We will carefully review your manuscript upon resubmission and further consideration of the manuscript is dependent on the text overlap being addressed in full. Please ensure that your revision is thorough as failure to address the concerns to our satisfaction may result in your submission not being considered further.

Reviewers' comments:

Reviewer's Responses to Questions

**Comments to the Author**

1. Does the manuscript provide a valid rationale for the proposed study, with clearly identified and justified research questions?

Reviewer #1: Partly

Reviewer #2: Yes

2. Is the protocol technically sound and planned in a manner that will lead to a meaningful outcome and allow testing the stated hypotheses?

Reviewer #1: Partly

Reviewer #2: Yes

3. Is the methodology feasible and described in sufficient detail to allow the work to be replicable?

Reviewer #1: No

Reviewer #2: Yes

4. Have the authors described where all data underlying the findings will be made available when the study is complete?

Reviewer #1: Yes

Reviewer #2: No

5. Is the manuscript presented in an intelligible fashion and written in standard English?

Reviewer #1: Yes

Reviewer #2: Yes

6. Review Comments to the Author

You may also provide optional suggestions and comments to authors that they might find helpful in planning their study.

Reviewer #1: Thank you for the read on an important topic. I am currently teaching social work students and many of them are interested in this! Overall good language and written with clearly in-depth knowledge and passion for the topic.

I would suggest the following as improvements before publishing:

My major overall concern while reading is that the scope of the intended review is very wide. To get an overview of every intervention and try to compare them seems almost unfeasible. While I applaud the obvious advantages of trying to cover both qualitative and quantitative papers, it may in fact be better if you were to split this into two separate sister papers, "What do we know about NBI, part I" and II. If you intend to integrate both qualitative and quantitative data into all the 5 parts of the analysis I fear the result will be unwieldy. Alternatively the aims outlined ca. line 172 could be cut to 2 or 3, not 5. Without this I fear the protocol will not be replicable.

INTRO

Thorough but maybe a bit disorganised -could be written with a more clear narrative and also use subheadings if allowed by Plos. For instance, small changes like a paragraph change in line 153 would help greatly on readability.

In line 160 you probably do not mean "alternatively" but maybe "in contrast" or something along those lines.

Readability could also be improved by shortcutting phrases like "It is anticipated that the qualitative analysis will capture..." -you could instead write "Our qualitative analysis will attempt to capture" -in general avoid the passive form throughout.The sentence in question is in general hard to follow and could probably be split in two or three.

Look for potential improvements like the above throughout.

In addition to the organisational points I am also missing a critical voice -there could be other reasons NBI are trending right now, e.g. that it is indeed a trend to be "natural", and that the increased interest does not in fact reflect actual effect. It would be good if the intro acknowledged this and made explicit an intent to include papers overtly critical to the concept of nature based interventions. And I would like to see that you are aware that such papers may to be missing from the reviews you are reviewing, and that you have considered what to do if this is the case.

I would also argue that the rise seen in interest in NBI may be partially economically motivated, and a part of the realisation on the part of western welfare states that institutions are very expensive to run, while self-help walking groups are much much cheaper. In other words I would wish for the intro to include a brief yet more nuanced and perhaps wider metaperspective on the rise of NBIs.

METHODS

Prisma and Prospero -good!

The "Data collection and verification" section of the paper could be much improved by providing a visual representation (table) of the extracted data plan you outline.

You say "Exact methods of synthesis for the included qualitative research will depend on the nature of the evidence identified" - in my opinion this would need to be refined quite a bit before publishing as it is too vague. Especially with regards to replicability, this would be essential in case someone else wants to use your methods section as a blueprint for a similar project.

Ethics: I think whenever you are going to be investigating qualitative data, it would be pertinent to include your own pre-existing potential biases, especially given that you are most likely planning to do a narrative review.

Reviewer #2: 1. Consider extending the searches beyond May 2022

2. Aim of the umbrella review lacks clarity in lines 153-155-Revise

3. The overall objective in lines 171-175 is convoluted and confusing to the reader as it includes the aim. It also does not speak to the aim outlined in lines 153-155-Revise

4. Specify the methodology being referred to in line 190-191

5. Definition of “umbrella review” in lines 193-195 should come earlier in the introduction section

6. The following statement in line 203 sounds incomplete: “No date limit will be placed on the search until.”

7. Justify why search will be limited to peer-reviewed studies published in English only-lines 230-231-This will make you leave out articles written in other languages

8. Justify why unpublished grey literature will not be included in the review

9. Definition of NBIs presented in lines 271-273 should come earlier in the introduction section

10. Avoid unnecessary repetitions e.g lines 275-279 are a repletion of lines 71-75

11. Consider utilizing a PRIMA flow diagram as part of Methods to indicate who the selection of studies will be done

12. Provide adequate details about how statistical and clinical heterogeneity will be determined and how this will affect data analysis

13. Manuscript will benefit from grammar and language editing

7. PLOS authors have the option to publish the peer review history of their article (what does this mean?). If published, this will include your full peer review and any attached files.

Reviewer #1: No

Reviewer #2: **Yes: **Genesis Chorwe-Sungani, PhD

---

## [Author Response · Author response to Decision Letter 0]

25 Jan 2023

Response to Editor and Reviewers Comments 

Dear Editor,

We would like to extend our sincerest thanks to you and the reviewers for taking the time to evaluate our manuscript and for allowing us to revise our study. We are now submitting our revised manuscript with tracked changes and a point-by-point response to the editor and both reviewers, which can be found below.

Academic Editor’s comments: 

Thank you to the editors for this guidance. We have formatted our manuscript in line with PLOS ONE’s style guidelines.

2. During our internal evaluation of the manuscript, we found significant text overlap throughout the whole manuscript between your submission and previous work .

We would like to make you aware that copying extracts from previous publications, especially outside the methods section, word-for-word, is unacceptable. In addition, the reproduction of text from published reports has implications for the copyright that may apply to the publications.

Please revise the manuscript to rephrase the duplicated text, cite your sources, and provide details as to how the current manuscript advances on previous work. Please note that further consideration is dependent on the submission of a manuscript that addresses these concerns about the overlap in text with published work.

We will carefully review your manuscript upon resubmission and further consideration of the manuscript is dependent on the text overlap being addressed in full. Please ensure that your revision is thorough as failure to address the concerns to our satisfaction may result in your submission not being considered further.

Thank you for bringing our attention to this. Academic integrity is central to our practice, and we would never knowingly take from others work. We have addressed this concern by using Turnitin and Grammarly software to detect any text overlap. All sources have been correctly cited and we provide clear reference to how the current manuscript advances on previous work. In our check, we found that most of the overlapping was with the preprint the submitted manuscript which has been published with medrxiv: https://www.medrxiv.org/. No further considerable overlaps were identified by us. Please do not hesitate to let us know if further issues with originality arise. 

Thank you for this point. This is a protocol for a systematic umbrella review, thus no empirical data has been collected to prepare this manuscript. We apologize for any errors in the Data availability statement. A note about this has been made in the cover letter. 

Reviewer #1

Thank you for the read on an important topic. I am currently teaching social work students and many of them are interested in this! Overall good language and written with clearly in-depth knowledge and passion for the topic.

I would suggest the following as improvements before publishing:

My major overall concern while reading is that the scope of the intended review is very wide. To get an overview of every intervention and try to compare them seems almost unfeasible. While I applaud the obvious advantages of trying to cover both qualitative and quantitative papers, it may in fact be better if you were to split this into two separate sister papers, "What do we know about NBI, part I" and II. If you intend to integrate both qualitative and quantitative data into all the 5 parts of the analysis I fear the result will be unwieldy. Alternatively the aims outlined ca. line 172 could be cut to 2 or 3, not 5. Without this I fear the protocol will not be replicable.

We would like to extend thanks to R1 for proposing the idea of two separate sister papers - separated into quantitative and qualitative evidence. We recognise your concerns regarding the scope of our review and objectives. We see significant merit in this idea provided the evidence would support such a divide. We have currently completed full-text screening and we are in the process of developing and piloting our data extraction sheet. By piloting the extraction sheet we aim to gauge if there is scope to divide the review into quantitative and qualitative evidence i.e. are there enough quantitative and qualitative reviews to warrant this and do the mixed-reviews present their findings in a way that would facilitate the separation of quantitative and qualitative evidence. We have added this stipulation to the protocol. See lines 620-628, p.19. Additionally we have addressed your concerns about the scope of the review being too wide by integrating objectives 3 and 4 into one (what are the factors that drive or limit the effectiveness of NBIs), and removing objective 5 but keep it as the ultimate goal of our review. See lines 281-289, p. 9

INTRO

Thorough but maybe a bit disorganised -could be written with a more clear narrative and also use subheadings if allowed by Plos. For instance, small changes like a paragraph change in line 153 would help greatly on readability.

We have organised the introduction to provide a more clear narrative of our study. Additionally, we have incorporated suggestions such as adding subheadings and paragraph changes. See introduction.

In line 160 you probably do not mean "alternatively" but maybe "in contrast" or something along those lines.

This has been amended. See line 249.

Readability could also be improved by shortcutting phrases like "It is anticipated that the qualitative analysis will capture..." -you could instead write "Our qualitative analysis will attempt to capture" -in general, avoid the passive form throughout. The sentence in question is in general hard to follow and could probably be split in two or three.

Look for potential improvements like the above throughout.

This particular example has been edited (See line 249-250) and we have altered similar instances throughout the manuscript where the passive form is used.

In addition to the organisational points I am also missing a critical voice -there could be other reasons NBI are trending right now, e.g. that it is indeed a trend to be "natural", and that the increased interest does not in fact reflect actual effect. It would be good if the intro acknowledged this and made explicit an intent to include papers overtly critical to the concept of nature based interventions. And I would like to see that you are aware that such papers may to be missing from the reviews you are reviewing and that you have considered what to do if this is the case.

I would also argue that the rise seen in interest in NBI may be partially economically motivated, and a part of the realisation on the part of western welfare states that institutions are very expensive to run, while self-help walking groups are much cheaper. In other words, I would wish for the intro to include a brief yet more nuanced and perhaps wider metaperspective on the rise of NBIs.

Thank you for this suggestion. We have provided a nuanced metaperspective on the rise of NBIs as suggested. We have also provided a more critical perspective on NBIs. See lines 142-174, p.5.

METHODS

Prisma and Prospero -good!

The "Data collection and verification" section of the paper could be much improved by providing a visual representation (table) of the extracted data plan you outline.

We have provided a visual representation (Table 1, p. 15-16) to represent our data extraction plan. See lines

You say "Exact methods of synthesis for the included qualitative research will depend on the nature of the evidence identified" - in my opinion this would need to be refined quite a bit before publishing as it is too vague. Especially with regards to replicability, this would be essential in case someone else wants to use your methods section as a blueprint for a similar project.

We have refined this statement to provide more detail on our methods of synthesis for the included qualitative studies. See lines 566-568, p. 18. 

Ethics: I think whenever you are going to be investigating qualitative data, it would be pertinent to include your own pre-existing potential biases, especially given that you are most likely planning to do a narrative review. 

Thank you for raising this issue. We have added a section outlining our commitment to practice reflexivity throughout the study. We will critically examine our own role, assumptions, beliefs, pre-existing potential bias and impact on the data during all stages of the research process including: (a) formulation of the research questions (b) data collection and (c ) data analysis. We hope that this reflexive engagement by multiple analysts will enhance the quality of this research. See lines 612-618, p.19.

Reviewer #2

Thank you R2 for your helpful and constructive feedback. Below we outline our response to your comments and subsequent action taken.

1. Consider extending the searches beyond May 2022

The search will be extended beyond May 2022. No date limit will be placed on the search. See line 322, p.10.

2. Aim of the umbrella review lacks clarity in lines 153-155-Revise

We have clarified our aim by adding more specific detail to this explanation. See lines 189-194.

3. The overall objective in lines 171-175 is convoluted and confusing to the reader as it includes the aim. 

It also does not speak to the aim outlined in lines 153-155-Revise

We have edited this to make sure that our overall objective links to the aim mentioned earlier in the introduction. Additionally, we have integrated objectives 3 and 4 into one (what are the factors that drive or limit the effectiveness of NBIs), and removed objective 5 but keep it as the ultimate goal of our review. We hope that this has provided a more concise overview of our specific objectives. See lines 237—289, p.9.

4. Specify the methodology being referred to in line 190-191

The specific methodology we were referring to has been outlined. See lines 246-247.

5. Definition of “umbrella review” in lines 193-195 should come earlier in the introduction section

Definition has been moved to earlier in the introduction section. See lines 200-202, p.7.

6. The following statement in line 203 sounds incomplete: “No date limit will be placed on the search until.”

This has been corrected. See line 322.

7. Justify why search will be limited to peer-reviewed studies published in English only-lines 230-231-This will make you leave out articles written in other languages

Our decision to include only peer-reviewed studies is based on our commitment to provide a synthesis of high quality evidence which has gone through a meticulous and rigorous review process. Our decision to include only studies published in English is the result of limited resources and the language constraints of our review team. While we acknowledge that this is a limitation of our review, including studies published in non-English languages would have increased resource challenges in relation to time, costs, and expertise in non-English languages. We have clarified this in the manuscript. See lines 357-363, p.11.

8. Justify why unpublished grey literature will not be included in the review

This decision is linked to our aforementioned commitment to include only studies which have gone through a rigorous review process which impacts upon study quality. This has been clarified (Aromataris et al., 2015) https://journals.lww.com/ijebh/Fulltext/2015/09000/Summarizing_systematic_reviews__methodological.4.aspx See lines 357-363.

9. Definition of NBIs presented in lines 271-273 should come earlier in the introduction section

This definition is now in the introduction section. See lines 311-312.

10. Avoid unnecessary repetitions e.g lines 275-279 are a repletion of lines 71-75

This repetition has been removed.

11. Consider utilizing a PRIMA flow diagram as part of Methods to indicate who the selection of studies will be done

We thank the Reviewer for this suggestion. We have made more evident our approach to data selection at p.12, where we have clarified the following: 

 “A PRISMA flow chart will be developed to record the screening and selection of studies. Once all records from our search are collected, EndNote 20 software (Thomson Reuters, Toronto, Ontario, Canada) will be used to remove duplicates and screen literature. A 2-stage screening process will be completed independently by two researchers (TS and CVYC); the first screening stage will consider titles and abstracts, while full-texts will be checked in the second stage”

A completed PRISMA flow diagram has not been included in the protocol as we would not be able to populate it until screening is completed. 

12. Provide adequate details about how statistical and clinical heterogeneity will be determined and how this will affect data analysis

We have clarified in the Data Analysis section (p.18, line 556-560) that: 

“Where possible and appropriate, statistical heterogeneity will be assessed following the Crochane 6.3 guidance (Deeks et al., 2022). Should trials of clinical relevance be included, an assessment of clinical heterogeneity, considering variability in participants, outcomes and characteristics of the intervention, will be assessed following guidance by Gagnier et al. (2013).”

13. Manuscript will benefit from grammar and language editing

Our review team has assessed the text for any grammar or language issues. Additionally, we have used specific software i.e. Grammarly to make changes to our grammar and language throughout the manuscript.

---

## [Decision Letter · Decision Letter 1]

15 May 2023

PONE-D-22-21700R1Factors influencing the effectiveness of nature–based Interventions (NBIs) aimed at improving mental health and wellbeing: Protocol of an umbrella reviewPLOS ONE

Dear Dr. Topaz,

Thank you for submitting your manuscript to PLOS ONE. After careful consideration, we feel that it has merit but does not fully meet PLOS ONE’s publication criteria as it currently stands. Therefore, we invite you to submit a revised version of the manuscript that addresses the points raised during the review process.

Please address the comments of reviewers. Please submit your revised manuscript by Jun 29 2023 11:59PM. If you will need more time than this to complete your revisions, please reply to this message or contact the journal office at plosone@plos.org. Please include the following items when submitting your revised manuscript:A rebuttal letter that responds to each point raised by the academic editor and reviewer(s). You should upload this letter as a separate file labeled 'Response to Reviewers'.A marked-up copy of your manuscript that highlights changes made to the original version. You should upload this as a separate file labeled 'Revised Manuscript with Track Changes'.An unmarked version of your revised paper without tracked changes. You should upload this as a separate file labeled 'Manuscript'.If applicable, we recommend that you deposit your laboratory protocols in protocols.io to enhance the reproducibility of your results. Protocols.io assigns your protocol its own identifier (DOI) so that it can be cited independently in the future. For instructions see: https://journals.plos.org/plosone/s/submission-guidelines#loc-laboratory-protocols. Additionally, PLOS ONE offers an option for publishing peer-reviewed Lab Protocol articles, which describe protocols hosted on protocols.io. Read more information on sharing protocols at https://plos.org/protocols?utm_medium=editorial-email&utm_source=authorletters&utm_campaign=protocols.

We look forward to receiving your revised manuscript.

Kind regards,

Md. Nazmul Huda, PhD

Academic Editor

PLOS ONE

Journal Requirements:

Reviewers' comments:

Reviewer's Responses to Questions

**Comments to the Author**

1. Does the manuscript provide a valid rationale for the proposed study, with clearly identified and justified research questions?

Reviewer #2: Yes

Reviewer #3: Yes

Reviewer #4: Yes

Reviewer #5: Yes

Reviewer #6: Partly

2. Is the protocol technically sound and planned in a manner that will lead to a meaningful outcome and allow testing the stated hypotheses?

Reviewer #2: Yes

Reviewer #3: Yes

Reviewer #4: Partly

Reviewer #5: Yes

Reviewer #6: Yes

3. Is the methodology feasible and described in sufficient detail to allow the work to be replicable?

Reviewer #2: Yes

Reviewer #3: Yes

Reviewer #4: Yes

Reviewer #5: Yes

Reviewer #6: Yes

4. Have the authors described where all data underlying the findings will be made available when the study is complete?

Reviewer #2: Yes

Reviewer #3: Yes

Reviewer #4: No

Reviewer #5: Yes

Reviewer #6: Yes

5. Is the manuscript presented in an intelligible fashion and written in standard English?

Reviewer #2: Yes

Reviewer #3: Yes

Reviewer #4: Yes

Reviewer #5: No

Reviewer #6: Yes

6. Review Comments to the Author

You may also provide optional suggestions and comments to authors that they might find helpful in planning their study.

Reviewer #2: The paper can be accepted for publication though responses for comment number 7 and 8 appears to be not adequate

Reviewer #3: Dear Author

Thank you for the opportunity to review your manuscript again. I appreciate the efforts you have made to address the previous reviewer's comments. Overall, the manuscript is well-written, and the methodology is sound. However, I still have a few observations which are mostly positive in nature.

Firstly, I agree that an umbrella review is necessary, given the number of systematic reviews on this topic. The authors' approach of only working with systematic reviews seems to be appropriate, but I am concerned about the vastness of the topic. Please ensure that the scope of the review is clearly defined, and the search strategy is comprehensive.

Secondly, I have no concerns with the methodology of the mixed-method review described in the protocol. The authors have provided sufficient explanation for the end date and English language criteria. I appreciate the thoroughness of the methodology.

Thirdly, while the introduction is well-structured and easy to follow now, I would have preferred a few examples of NBI in addition to the definition(s). This would help readers like me to better understand the concept.

Fourthly, since most of the concerns raised by the previous reviewer have been addressed, I find the protocol to be robust. However, there is undue repetition in the 'Intervention' section. Specifically, line 312-316 and 318-321 are exact copies of line 94-98 and 100-105 from the 'Introduction' section. Please avoid such repetition and try to explain it differently in the 'Intervention' section to make it clear to the reader. By differently, I mean use the space in the intervention section to expand on what you said in the Intro section.

In summary, the manuscript is well-written, and the methodology is sound. My observations are mostly positive, and I appreciate the authors' efforts to address the previous reviewer's comments. Please consider my feedback while revising the protocol.

Reviewer #4: 1/ good topics but its too abroad better to be restricted to specific area will be more informative

2/ if the time extended beyond that period 2022 will b better

Reviewer #5: 1. References : some of the references are outdated . authors can chose the recent 5 years papers.

2. Authors should simply the text and focus on grammar more.

Reviewer #6: Dear Editor,

Thank you for sending me a manuscript of study Protocol PONE-D-22-21700R1 for reviewing titled ‘Factors influencing the effectiveness of nature–based Interventions (NBIs) aimed at improving mental health and wellbeing: Protocol of an umbrella review’.

Since this manuscript has already been reviewed by another reviewer earlier, it was of ample advantages for me to go for reviewing for the second time.

The very title: ‘Factors influencing the effectiveness of nature–based Interventions (NBIs) aimed at improving mental health and wellbeing: Protocol of an umbrella review’ – Loos okay and self-explanatory.

To start with:

The abstract seems well described in a consiced yet in a meaningful way. Alike other systematic reviews supporting NBIs as a mechanism of enhancing mental health and wellbeing, the authors claimed that available evidence for the effectiveness of NBIs remain fragmentary and mixed that yields significant fragmentation of knowledge within the field making it difficult to draw firm conclusions on an NBI.

Aim(s) and Objective(s):

The authors, aimed to study this mixed method umbrella review by synthesizing evidence on the effectiveness of NBIs as a summative review of available published systematic reviews and meta-analyses. The authors conducted a systematic search using 13 search engines, like: PsycINFO, PubMed, Greenfile, Web of Science, Embase, Scopus, Academic Search Complete (EBSCO), Environment Complete (EBSCO), Cochrane Library, CINAHL, Health Policy Reference Center and Google Scholar for a period of its inception (as the authors claimed) up to May 2022.

Methodological drives:

Strategically, the authors’ aimed to find out all (not mentioned though) published systematic reviews of NBIs yielding improved health and wellbeing as their explicit goal.

Then, for synthesis of this systematic reviews they used a mixed method (quantitative & qualitative data) engaging two independent authors who did the following steps essential for a modest review:

- Literature search,

- Record screening,

- Data extracting, and then,

- Quality assessment of each of all (not clear) systematic review and meta-analysis.

- The authors synthesized all individually qualitative & quantitative syntheses parallelly but then combined those in an overarching narrative synthesis and used the quantitative evidence to assess strength and direction of effect of NBIs on outcome of mental health and wellbeing.

Yielded results and findings:

- The authors analyzed evidences drawn from qualitative studies and synthesized to those to understand various pathways to engagement, process of involvement and experiential factors which may have mediated experiences.

- However, the authors assessed the calculated risk of bias of systematic reviews will be using a 16-item Assessment of Multiple Systematic Reviews 2 (AMSTAR2) checklist

- Finally, registered on international database of prospectively registered systematic reviews in health and social care- PROSPERO (CRD42022329179) to record & maintain that as a permanent record.

My comment on the response of authors to earlier two reviewers including reply to editor’s earlier comments, are as below:

Now, that this manuscript has been reviewed by two others including the editorial input I can guess that the state of this manuscript currently looks great as a post edited copy. However, followings remain my final comment on this pre-reviewed manuscript, par se.

Authors 1st reply to editorial review/comment earlier:

I am glad to notice that the authors revised the manuscript with point specific answers to the editor using tracked changes method.

The five Responses by the authors to each of the Academic Editor’s comments (Powered by Editorial Manager® and ProduXion Manager® from Aries Systems Corp.) remains acceptable and good to notice that accepting all the 5 comments/ queries that were raised by the editor, the authors worked on those points seriously and thus replied modestly yet logically, pointing out the corrections they made as follows:

1.The authors ensured that their manuscript met PLOS ONE's style requirements and file naming.

2.The authors agreed, attended and corrected all the error to correct those as per editor’s advices what they found during internal evaluation of the manuscript.

3.So, now it is the time for the editor to carefully review the manuscript finally that they resubmitted. But, to me it now looks okay.

4.To reply to the editor’s query to ensure by the authors that their revision is thorough so it can be acceptable in this stage, I think.

5.Regarding R-1 comment on providing repository information for author’s data, I think the authors reply is to be validated by the editor’s office yet, if it remains acceptable what the author’s pointed out on their protocol (a systematic umbrella review) may not have any empirical data to be used in preparing this manuscript, and, so the authors apologized for any errors in the data availability statement including a note on this has been reflected in their cover letter.

Next, looking at the critically raised comments by Reviewer #1 and the replies by the authors also though seems to me as acceptable but it entirely depends on R-1 and the editorial board to re-examining the authors reactions, opinions and replies, if acceptable.

However, the authors reply to this proposition that when they liked the proposed idea by R-1 of preparing two separate sister papers analyzing quantitative & qualitative evidences, separately, but explained that the scope of their review & objectives was not really the same but they had significant merit in Powered by Editorial Manager® and ProduXion Manager® from Aries Systems Corporation this idea provided the evidence would support such a divide. And, that the authors informed that they have completed full-text screening and on the process of developing/piloting their data extraction. By piloting the extraction sheet, they aim to gauge if there is any scope to divide the review into quantitative and qualitative evidence. However, the authors have added this stipulation to the protocol on lines 620-628, on page 19.

Moreover, to reply R-1’s concern on the scope of review being too wide by integrating objectives 3 and 4 into one …. the authors replied that they have organized the introduction to provide a clearer narrative of the study. Additionally, they have incorporated suggestions such as adding subheadings and paragraph changes. ……. However, the authors has edited (See line 249-250) and altered similar instances throughout the manuscript where the passive form is used.

In addition to the organizational points from R-1 point of view, in missing out a critical voice -there could be other reasons NBI are trending right now, e.g. that it is indeed a trend to be "natural", and that the increased interest does not in fact reflect actual effect….etc.., the authors replied that they have also provided a nuanced meta-perspective on the rise of NBIs that the R-1 suggested and they also provided a more critical perspective on NBIs. See lines 142-174, p.5.

METHODS Prisma and Prospero -good! …… The "Data collection and verification" section of the paper could be much improved by providing a visual representation (table) of the extracted data plan outlined……… … etc. The authors already refined this statement to provide more detail on our methods of synthesis for the included qualitative studies. See lines 566-568, p. 18.

On the point that R-1 raised on Ethics- the authors have added a section outlining their commitment to practice reflexivity throughout the study. And they said that they will critically examine our own role, assumptions, beliefs, pre-existing potential bias and impact on the data during all stages of the research process including: (a) formulation of the research questions, (b) data collection and (c) data analysis. Moreover, they hope that reflexive engagement by multiple analysts will enhance the quality of this research. See lines 612-618, p.19.

Thanking the Reviewer #2 for helpful and constructive feedback the authors outlined their response to R-2 comments and action taken subsequently. Moreover, as per R-2 comment the authors agreed and considered extending searches beyond May 2022.

Then, to reply to a valid point that R#2 raised on ‘Aim of the umbrella review lacks clarity in lines 153-155’ suggested for revising, the authors agreed to R-2 to clarify the aim by adding more specific detail to this explanation. See lines 189-194.

Again, the overall objective in lines 171-175 was convoluted and confusing to the reader as R#2 commented, as it includes the aim and also does not speak to the aim outlined in lines 153-155-Revise. So, the authors have edited this to make sure that our overall objective links to the aim mentioned earlier in the introduction.

Additionally, the authors integrated objectives 3 and 4 into one (what are the factors that drive or limit the effectiveness of NBIs), and removed objective 5 but kept it as the ultimate goal. And, the authors provided a more concise overview on the specific objectives (lines 237—289, p.9. 4.) And the authors mentioned that the specific methodology they referring to was outlined. See lines 246-247.

Definition of “umbrella review” in lines 193-195 should come earlier in the introduction section Definition has been moved to earlier in the introduction section. See lines 200-202, p.7. 6. To answer to the R#2 statement in line 203 that sounded incomplete: “No date limit will be placed on the search until.”

This has been corrected. See line 322. 7. Justify why search will be limited to peer-reviewed studies published in English only lines 230-231- This will make you leave out articles written in other languages., the authors replied as follows:

‘’Our decision to include only peer-reviewed studies is based on our commitment to provide a synthesis of high-quality evidence which has gone through a meticulous and rigorous review process. Our decision to include only studies published in English is the result of limited resources and the language constraints of our review team. While we acknowledge that this is a limitation of our review, including studies published in Powered by Editorial Manager® and ProduXion Manager® from Aries Systems Corporation non-English languages would have increased resource challenges in relation to time, costs, and expertise in non-English languages.

We have clarified this in the manuscript. See lines 357-363, p.11.’’

The authors have cited the ‘definition’ in introduction section (lines 311-312) as the R#2 suggested.

Also, according to R#2 suggestion the authors removed all unnecessary repetitions.

The authors thankfully accepted the good suggestion by R#2 to consider utilizing a PRIMA flow diagram as part of methods to indicate who the selection of studies were done, and thus, the authors made their approach more evident to data selection at p.12, where they have clarified the following: “A PRISMA flow chart will be developed to record the screening and selection of studies. Once all records from our search are collected, EndNote 20 software (Thomson Reuters, Toronto, Ontario, Canada) will be used to remove duplicates and screen literature. A 2-stage screening process will be completed independently by two researchers (TS and CVYC); the first screening stage will consider titles and abstracts, while full-texts will be checked in the second stage” A completed PRISMA flow diagram has not been included in the protocol as we would not be able to populate it until screening is completed.

And, according to R#2 suggestion the authors provided adequate details on how statistical and clinical heterogeneity will be determined and how this will affect data analysis (clarified in Data Analysis section on p.18, line 556-560. Additionally, the authors have used specific software i.e. Grammarly to make changes to our grammar and language throughout the manuscript.

Finally, the bibliography:

To me the reference list remains quite updated and all the 53 citations were cited from recent literature.

However, my specific comment:

The authors must comply with checking the existing grammatical errors & spelling mistakes all through including improving the English language (better if edited by any native English-spoken person).

My last impression and final comment:

Gauging the depth of the research topic, quality of manuscript submitted (standard of research protocol: aim, objective, methodology and findings) and the outcome of the study if valuable and the avlues it might add in our scientific research bank, I do recommend that this manuscript be published in any of the recent issues of PLoS, provided all the suggested editing/corrections are reflected in the final manuscript before it is published finally.

Comment: Recommended for publication with minor revisions.

7. PLOS authors have the option to publish the peer review history of their article (what does this mean?). If published, this will include your full peer review and any attached files.

Reviewer #2: **Yes: **Genesis Chorwe-Sungani

Reviewer #3: No

Reviewer #4: No

Reviewer #5: **Yes: **Nadia Anwar

Reviewer #6: **Yes: **Dr. Kazi Selim Anwar, MD, MPhil (Engand), Head, Medicl Research Unit (MRU), Ad-din Women's Medcial College, Dhaka

---

## [Author Response · Author response to Decision Letter 1]

29 Jun 2023

Response to Editor and Reviewers Comments 

Dear Editor,

We would like to extend our sincerest thanks to you and the reviewers for taking the time to evaluate our revised manuscript with tracked changes and for providing us with feedback. We are grateful to you for allowing us to revise our study and incorporate these suggestions. We are now submitting our revised manuscript with tracked changes and a point-by-point response to the editor and both reviewers, which can be found below.

Reviewer #2: 

The paper can be accepted for publication though responses for comment number 7 and 8 appear to be not adequate

We are unsure what the reviewer is referring to in relation to comment number 7 and 8. If the reviewer is referring to our initial response (in the first stage of review) to reviewer 2’s comments 7 and 8 we would like to expand on our previous response to explain that the focus of our review is on systematic reviews and meta-analyses including a quality assessment, which is a typical publication found among peer-reviewed publications. We would be very grateful if reviewer 2 or the academic editor could let us know what is required of us to make it adequate. Thank you for your comments we have included our initial response below for ease of remembrance: 

“7. Justify why search will be limited to peer-reviewed studies published in English only-lines 230-231-This will make you leave out articles written in other languages

Our decision to include only peer-reviewed studies is based on our commitment to provide a synthesis of high-quality evidence which has gone through a meticulous and rigorous review process. Our decision to include only studies published in English is the result of limited resources and the language constraints of our review team. While we acknowledge that this is a limitation of our review, including studies published in non-English languages would have increased resource challenges in relation to time, costs, and expertise in non-English languages. We have clarified this in the manuscript. 

8. Justify why unpublished grey literature will not be included in the review

This decision is linked to our aforementioned commitment to include only studies which have gone through a rigorous review process which impacts study quality. This has been clarified (Aromataris et al., 2015) https://journals.lww.com/ijebh/Fulltext/2015/09000/Summarizing_systematic_reviews__methodological.4.aspx See lines 357-363.”

Reviewer #3: 

Dear Author

Thank you for the opportunity to review your manuscript again. I appreciate the efforts you have made to address the previous reviewer's comments. Overall, the manuscript is well-written, and the methodology is sound. However, I still have a few observations which are mostly positive in nature.

Firstly, I agree that an umbrella review is necessary, given the number of systematic reviews on this topic. The authors' approach of only working with systematic reviews seems to be appropriate, but I am concerned about the vastness of the topic. Please ensure that the scope of the review is clearly defined, and the search strategy is comprehensive.

We would like to thank you for recognising the need for this umbrella review. The scope of the review has been clearly defined in lines 172- 216 and the search strategy has been outlined in lines 215-248.

Secondly, I have no concerns with the methodology of the mixed-method review described in the protocol. The authors have provided sufficient explanation for the end date and English language criteria. I appreciate the thoroughness of the methodology.

Thirdly, while the introduction is well-structured and easy to follow now, I would have preferred a few examples of NBI in addition to the definition(s). This would help readers like me to better understand the concept.

We have added some examples of NBIs to the introduction as per your suggestion. See lines 81-86.

Fourthly, since most of the concerns raised by the previous reviewer have been addressed, I find the protocol to be robust. However, there is undue repetition in the 'Intervention' section. Specifically, line 312-316 and 318-321 are exact copies of line 94-98 and 100-105 from the 'Introduction' section. Please avoid such repetition and try to explain it differently in the 'Intervention' section to make it clear to the reader. By differently, I mean use the space in the intervention section to expand on what you said in the Intro section.

Thank you for highlighting this unnecessary repetition. We have removed all repetition and have changed the wording in the ‘Intervention’ section to expand on what is said in the Introduction. See lines 374-393.

In summary, the manuscript is well-written, and the methodology is sound. My observations are mostly positive, and I appreciate the authors' efforts to address the previous reviewer's comments. Please consider my feedback while revising the protocol.

Reviewer #4:

 1/ good topics but its too abroad better to be restricted to specific area will be more informative 2/ if the time extended beyond that period 2022 will be better

Thanks to reviewer 4 for highlighting the merit of this review. Regarding point 1, the broad scope is coherent with the need for an overall synthesis of the available evidence and using an umbrella review demonstrates that. As mentioned in our response to our initial reviewers in the first round of review, the search will be extended beyond May 2022. In relation to point 2, no date limit will be placed on the search.

Reviewer #5: 

References : some of the references are outdated . authors can chose the recent 5 years papers. 2. Authors should simply the text and focus on grammar more.

The reference list remains updated and all 53 citations were cited from recent literature.

We have simplified the text where possible and have used specific software i.e. Grammarly, Word Grammar and Spelling Editor to check the grammar of our manuscript. Additionally, all research team members have checked the manuscript for issues with grammar etc. 

Reviewer #6: 

Dear Editor,

Thank you for sending me a manuscript of study Protocol PONE-D-22-21700R1 for reviewing titled ‘Factors influencing the effectiveness of nature–based Interventions (NBIs) aimed at improving mental health and wellbeing: Protocol of an umbrella review’.

Since this manuscript has already been reviewed by another reviewer earlier, it was of ample advantages for me to go for reviewing for the second time.

The very title: ‘Factors influencing the effectiveness of nature–based Interventions (NBIs) aimed at improving mental health and wellbeing: Protocol of an umbrella review’ – Looks okay and self-explanatory.

To start with:

The abstract seems well described in a concise yet in a meaningful way. Alike other systematic reviews supporting NBIs as a mechanism of enhancing mental health and wellbeing, the authors claimed that available evidence for the effectiveness of NBIs remain fragmentary and mixed that yields significant fragmentation of knowledge within the field making it difficult to draw firm conclusions on an NBI.

Aim(s) and Objective(s):

The authors, aimed to study this mixed method umbrella review by synthesizing evidence on the effectiveness of NBIs as a summative review of available published systematic reviews and meta-analyses. The authors conducted a systematic search using 13 search engines, like: PsycINFO, PubMed, Greenfile, Web of Science, Embase, Scopus, Academic Search Complete (EBSCO), Environment Complete (EBSCO), Cochrane Library, CINAHL, Health Policy Reference Center and Google Scholar for a period of its inception (as the authors claimed) up to May 2022.

Methodological drives:

Strategically, the authors’ aimed to find out all (not mentioned though) published systematic reviews of NBIs yielding improved health and wellbeing as their explicit goal.

Then, for synthesis of this systematic reviews they used a mixed method (quantitative & qualitative data) engaging two independent authors who did the following steps essential for a modest review:

- Literature search,

- Record screening,

- Data extracting, and then,

- Quality assessment of each of all (not clear) systematic review and meta-analysis.

- The authors synthesized all individually qualitative & quantitative syntheses parallelly but then combined those in an overarching narrative synthesis and used the quantitative evidence to assess strength and direction of effect of NBIs on outcome of mental health and wellbeing.

Yielded results and findings:

- The authors analyzed evidences drawn from qualitative studies and synthesized to those to understand various pathways to engagement, process of involvement and experiential factors which may have mediated experiences.

- However, the authors assessed the calculated risk of bias of systematic reviews will be using a 16-item Assessment of Multiple Systematic Reviews 2 (AMSTAR2) checklist

- Finally, registered on international database of prospectively registered systematic reviews in health and social care- PROSPERO (CRD42022329179) to record & maintain that as a permanent record.

My comment on the response of authors to earlier two reviewers including reply to editor’s earlier comments, are as below:

Now, that this manuscript has been reviewed by two others including the editorial input I can guess that the state of this manuscript currently looks great as a post edited copy. However, followings remain my final comment on this pre-reviewed manuscript, par se.

Authors 1st reply to editorial review/comment earlier:

I am glad to notice that the authors revised the manuscript with point specific answers to the editor using tracked changes method.

The five Responses by the authors to each of the Academic Editor’s comments (Powered by Editorial Manager® and ProduXion Manager® from Aries Systems Corp.) remains acceptable and good to notice that accepting all the 5 comments/ queries that were raised by the editor, the authors worked on those points seriously and thus replied modestly yet logically, pointing out the corrections they made as follows:

1.The authors ensured that their manuscript met PLOS ONE's style requirements and file naming.

2.The authors agreed, attended and corrected all the error to correct those as per editor’s advices what they found during internal evaluation of the manuscript.

3.So, now it is the time for the editor to carefully review the manuscript finally that they resubmitted. But, to me it now looks okay.

4.To reply to the editor’s query to ensure by the authors that their revision is thorough so it can be acceptable in this stage, I think.

5.Regarding R-1 comment on providing repository information for author’s data, I think the authors reply is to be validated by the editor’s office yet, if it remains acceptable what the author’s pointed out on their protocol (a systematic umbrella review) may not have any empirical data to be used in preparing this manuscript, and, so the authors apologized for any errors in the data availability statement including a note on this has been reflected in their cover letter.

Next, looking at the critically raised comments by Reviewer #1 and the replies by the authors also though seems to me as acceptable but it entirely depends on R-1 and the editorial board to re-examining the authors reactions, opinions and replies, if acceptable.

However, the authors reply to this proposition that when they liked the proposed idea by R-1 of preparing two separate sister papers analyzing quantitative & qualitative evidences, separately, but explained that the scope of their review & objectives was not really the same but they had significant merit in Powered by Editorial Manager® and ProduXion Manager® from Aries Systems Corporation this idea provided the evidence would support such a divide. And, that the authors informed that they have completed full-text screening and on the process of developing/piloting their data extraction. By piloting the extraction sheet, they aim to gauge if there is any scope to divide the review into quantitative and qualitative evidence. However, the authors have added this stipulation to the protocol on lines 620-628, on page 19.

Moreover, to reply R-1’s concern on the scope of review being too wide by integrating objectives 3 and 4 into one …. the authors replied that they have organized the introduction to provide a clearer narrative of the study. Additionally, they have incorporated suggestions such as adding subheadings and paragraph changes. ……. However, the authors has edited (See line 249-250) and altered similar instances throughout the manuscript where the passive form is used.

In addition to the organizational points from R-1 point of view, in missing out a critical voice -there could be other reasons NBI are trending right now, e.g. that it is indeed a trend to be "natural", and that the increased interest does not in fact reflect actual effect….etc.., the authors replied that they have also provided a nuanced meta-perspective on the rise of NBIs that the R-1 suggested and they also provided a more critical perspective on NBIs. See lines 142-174, p.5.

METHODS Prisma and Prospero -good! …… The "Data collection and verification" section of the paper could be much improved by providing a visual representation (table) of the extracted data plan outlined……… … etc. The authors already refined this statement to provide more detail on our methods of synthesis for the included qualitative studies. See lines 566-568, p. 18.

On the point that R-1 raised on Ethics- the authors have added a section outlining their commitment to practice reflexivity throughout the study. And they said that they will critically examine our own role, assumptions, beliefs, pre-existing potential bias and impact on the data during all stages of the research process including: (a) formulation of the research questions, (b) data collection and (c) data analysis. Moreover, they hope that reflexive engagement by multiple analysts will enhance the quality of this research. See lines 612-618, p.19.

Thanking the Reviewer #2 for helpful and constructive feedback the authors outlined their response to R-2 comments and action taken subsequently. Moreover, as per R-2 comment the authors agreed and considered extending searches beyond May 2022.

Then, to reply to a valid point that R#2 raised on ‘Aim of the umbrella review lacks clarity in lines 153-155’ suggested for revising, the authors agreed to R-2 to clarify the aim by adding more specific detail to this explanation. See lines 189-194.

Again, the overall objective in lines 171-175 was convoluted and confusing to the reader as R#2 commented, as it includes the aim and also does not speak to the aim outlined in lines 153-155-Revise. So, the authors have edited this to make sure that our overall objective links to the aim mentioned earlier in the introduction.

Additionally, the authors integrated objectives 3 and 4 into one (what are the factors that drive or limit the effectiveness of NBIs), and removed objective 5 but kept it as the ultimate goal. And, the authors provided a more concise overview on the specific objectives (lines 237—289, p.9. 4.) And the authors mentioned that the specific methodology they referring to was outlined. See lines 246-247.

Definition of “umbrella review” in lines 193-195 should come earlier in the introduction section Definition has been moved to earlier in the introduction section. See lines 200-202, p.7. 6. To answer to the R#2 statement in line 203 that sounded incomplete: “No date limit will be placed on the search until.”

This has been corrected. See line 322. 7. Justify why search will be limited to peer-reviewed studies published in English only lines 230-231- This will make you leave out articles written in other languages., the authors replied as follows:

‘’Our decision to include only peer-reviewed studies is based on our commitment to provide a synthesis of high-quality evidence which has gone through a meticulous and rigorous review process. Our decision to include only studies published in English is the result of limited resources and the language constraints of our review team. While we acknowledge that this is a limitation of our review, including studies published in Powered by Editorial Manager® and ProduXion Manager® from Aries Systems Corporation non-English languages would have increased resource challenges in relation to time, costs, and expertise in non-English languages.

We have clarified this in the manuscript. See lines 357-363, p.11.’’

The authors have cited the ‘definition’ in introduction section (lines 311-312) as the R#2 suggested.

Also, according to R#2 suggestion the authors removed all unnecessary repetitions.

The authors thankfully accepted the good suggestion by R#2 to consider utilizing a PRIMA flow diagram as part of methods to indicate who the selection of studies were done, and thus, the authors made their approach more evident to data selection at p.12, where they have clarified the following: “A PRISMA flow chart will be developed to record the screening and selection of studies. Once all records from our search are collected, EndNote 20 software (Thomson Reuters, Toronto, Ontario, Canada) will be used to remove duplicates and screen literature. A 2-stage screening process will be completed independently by two researchers (TS and CVYC); the first screening stage will consider titles and abstracts, while full-texts will be checked in the second stage” A completed PRISMA flow diagram has not been included in the protocol as we would not be able to populate it until screening is completed.

And, according to R#2 suggestion the authors provided adequate details on how statistical and clinical heterogeneity will be determined and how this will affect data analysis (clarified in Data Analysis section on p.18, line 556-560. Additionally, the authors have used specific software i.e. Grammarly to make changes to our grammar and language throughout the manuscript.

We would like to thank you for taking the time to assess the revised manuscript and providing a detailed summary of the initial review process and our responses to the reviewer's suggestions. 

Finally, the bibliography:

To me the reference list remains quite updated and all the 53 citations were cited from recent literature.

Thank you for acknowledging this re the bibliography.

However, my specific comment:

The authors must comply with checking the existing grammatical errors & spelling mistakes all through including improving the English language (better if edited by any native English-spoken person).

The research team (comprised of 4 individuals and 3 native English speakers) have checked existing grammatical and spelling errors. Additionally, we have used the appropriate software Grammarly to double-check these issues.

My last impression and final comment:

Gauging the depth of the research topic, quality of manuscript submitted (standard of research protocol: aim, objective, methodology and findings) and the outcome of the study if valuable and the avlues it might add in our scientific research bank, I do recommend that this manuscript be published in any of the recent issues of PLoS, provided all the suggested editing/corrections are reflected in the final manuscript before it is published finally.

Comment: Recommended for publication with minor revisions.

Thank you to the academic editor and reviewers for taking the time to look at our manuscript and providing such thorough feedback. We hope that you agree our manuscript is much improved and consider it ready for publication.

---

## [Editor Report · Decision Letter 2]

2 Jul 2023

Factors influencing the effectiveness of nature–based Interventions (NBIs) aimed at improving mental health and wellbeing: Protocol of an umbrella review

PONE-D-22-21700R2

Dear Dr. Topaz Shrestha,

We’re pleased to inform you that your manuscript has been judged scientifically suitable for publication and will be formally accepted for publication once it meets all outstanding technical requirements.

Kind regards,

Md. Nazmul Huda, PhD

Academic Editor

PLOS ONE

Additional Editor Comments (optional):

Dear author,

I think you addressed the comments. Therefore, we do not recommend further reviews of this manuscript. Thanks.

Nazmul
---

## [Editor Report · Acceptance letter]

12 Jul 2023

PONE-D-22-21700R2 

Factors influencing the effectiveness of Nature-based Interventions (NBIs) aimed at improving mental health and wellbeing: Protocol of an umbrella review 

Dear Dr. Shrestha:

I'm pleased to inform you that your manuscript has been deemed suitable for publication in PLOS ONE. Congratulations! Your manuscript is now with our production department. 

Kind regards, 

on behalf of

Dr. Md. Nazmul Huda 

Academic Editor

PLOS ONE